# Localized semi-nonnegative matrix factorization (LocaNMF) of widefield calcium imaging data

Shreya Saxena[1,2,3,4]*, Ian Kinsella[1,2,3], Simon Musall[5], Sharon H. Kim[1,6],
Jozsef Meszaros[1,6], David N. Thibodeaux[1,6], Carla Kim[1,6], John Cunningham[1,2,3,4],
Elizabeth M. C. Hillman[1,6], Anne Churchland[5], Liam Paninski[1,2,3,4,7]

1 Mortimer B. Zuckerman Mind Brain Behavior Institute, Columbia University, New York, New York, United States of America, 2 Department of Statistics, Columbia University, New York, New York, United States of America, 3 Center for Theoretical Neuroscience, Columbia University, New York, New York, United States of America, 4 Grossman Center for the Statistics of Mind, Columbia University, New York, New York, United States of America, 5 Cold Spring Harbor Laboratory, Cold Spring Harbor, New York, United States of America, 6 Laboratory for Functional Optical Imaging, Department of Biomedical Engineering, Columbia University, New York, New York, United States of America, 7 Department of Neuroscience, Columbia University, New York, New York, United States of America

* ss5513@columbia.edu

**Data Availability Statement:** The majority of the data from this manuscript is publicly available at the following location: http://repository.cshl.edu/id/eprint/38599/.

## Abstract

Widefield calcium imaging enables recording of large-scale neural activity across the mouse dorsal cortex. In order to examine the relationship of these neural signals to the resulting behavior, it is critical to demix the recordings into meaningful spatial and temporal components that can be mapped onto well-defined brain regions. However, no current tools satisfactorily extract the activity of the different brain regions in individual mice in a data-driven manner, while taking into account mouse-specific and preparation-specific differences. Here, we introduce Localized semi-Nonnegative Matrix Factorization (LocaNMF), a method that efficiently decomposes widefield video data and allows us to directly compare activity across multiple mice by outputting mouse-specific localized functional regions that are significantly more interpretable than more traditional decomposition techniques. Moreover, it provides a natural subspace to directly compare correlation maps and neural dynamics across different behaviors, mice, and experimental conditions, and enables identification of task- and movement-related brain regions.

## Author summary

While recording from multiple regions of the brain, how does one best incorporate prior information about anatomical regions while accurately representing the data? Here, we introduce Localized semi-NMF (LocaNMF), an algorithm that efficiently decomposes widefield video data into meaningful spatial and temporal components that can be decoded and compared across different behavioral sessions and experimental conditions. Mapping the inferred components onto well-defined brain regions using a widely-used brain atlas provides an interpretable, stable decomposition. LocaNMF allows us to

**Funding:** We gratefully acknowledge support from the Swiss National Science Foundation P2SKP2_178197 (SS), P300PB_174369 (SM), NIBIB R01 EB22913 (LP), the Simons Foundation via the International Brain Lab collaboration (LP, AC), NSF Neuronex DBI-1707398 (LP), NIH/NINDS 1U19NS104649-01 (LP, EH), NIH/NIMH 1 RF1 MH114276-01 (EH), NIH/NINDS 1R01NS063226-08 (EH), NIH/EY R01EY022979 (AC), and Columbia University's Research Opportunities and Approaches to Data Science program (EH). The funders had no role in study design, data collection and analysis, decision to publish, or preparation of the manuscript.

**Competing interests:** The authors have declared that no competing interests exist.

satisfactorily extract the activity of the different brain regions in individual mice in a data-driven manner, while taking into account mouse-specific and preparation-specific differences.

This is a *PLOS Computational Biology* Methods paper.

## Introduction

A fundamental goal in neuroscience is to simultaneously record from as many neurons as possible, with high temporal and spatial resolution [1]. Unfortunately, tradeoffs must be made: high-resolution recording methods often lead to small fields of view, and vice versa. Widefield calcium imaging (WFCI) methods provide a compromise: this approach offers a global view of the (superficial) dorsal cortex, with temporal resolution limited only by the activity indicator, calcium dynamics and camera speeds. Single-cell resolution of superficial neurons is possible using a "crystal skull" preparation [2] but simpler, less invasive thinned-skull preparations that provide spatial resolution of around tens of microns per pixel have become increasingly popular [2–14]; of course there is also a large relevant literature on widefield voltage and intrinsic signal imaging [15–18].

How should we approach the analysis of WFCI data? In the context of single-cell-resolution data, the basic problems are clear: we want to *denoise* the CI video data, *demix* this data into signals from individual neurons, and then in many cases it is desirable to *deconvolve* these signals to estimate the underlying activity of each individual neuron; see e.g. [19] and references therein for further discussion of these issues.

For data that lacks single-neuron resolution, the relevant analysis goals require further reflection. One important goal (regardless of spatial resolution) is to *compress* and *denoise* the large, noisy datasets resulting from WFCI experiments, to facilitate downstream analyses [20]. Another critical goal is to decompose the video into a collection of interpretable signals that capture all of the useful information in the dataset. What do we mean by "interpretable" here? Ideally, each signal we extract should be referenced to a well-defined region of the brain (or multiple regions)—but at the same time the decomposition approach should be flexible enough to adapt to anatomical differences across animals. The extracted signals should be comparable across animals performing the same behavioral task, or presented with the same sensory stimulus; at the very least the decomposition should be reproducible when computed on data collected from different comparable experimental blocks from the same animal.

Do existing analysis approaches satisfy these desiderata? One common approach is to define regions of interest (ROIs), either automatically or manually, and then to extract signals by averaging within ROIs [7]. However, this approach discards significant information outside the ROIs, and fails to demix multiple signals that may overlap spatially within a given ROI. Alternatively, we could apply principal components analysis (PCA), by computing the singular value decomposition (SVD) of the video [8]. The resulting principal components serve to decompose the video into spatial and temporal terms that can capture the majority of available signal in the dataset. However, these spatial components are typically de-localized (i.e., they have support over the majority of the field of view, instead of being localized to well-defined brain regions). In addition, the vectors output by SVD are constrained to be orthogonal by

construction, but there is no a priori reason to expect this orthogonality constraint to lead to more interpretable or reproducible extracted components. Indeed, in practice SVD-based components are typically not reproducible across recording sessions from the same animal: the PCs from one session may look very different from the PCs from another session (though the vector subspace spanned by these PCs may be similar across sessions). Non-negative matrix factorization (NMF) is a decomposition approach that optimizes a similar cost function as SVD, without orthogonality constraints but with additional non-negativity constraints on the spatial and/or temporal components [6, 21]; unfortunately, as we discuss below, many of the same criticisms of PCA also apply to NMF. Finally, seed-pixel correlation maps [7] provide a useful exploratory approach for visualizing the correlation structure in the data, but do not provide a meaningful decomposition of the full video into interpretable signals per se.

In this work we introduce a new approach to perform a localized, more interpretable decomposition of WFCI data. The proposed approach is a variation on classical NMF, termed localized semi-NMF (LocaNMF), that decomposes the widefield activity by (a) using existing brain atlases to initialize the estimated spatial components, and (b) limiting the spread of each spatial component in order to obtain localized components. We provide both CPU and GPU implementations of the algorithm in the code here. Running LocaNMF allows us to efficiently obtain temporal components localized to well-defined brain regions in a data-driven manner. Empirically, we find that the resulting components satisfy the reproducibility desiderata described above, leading to a more interpretable decomposition of WFCI data. In experimental data from mice expressing different calcium indicators and exhibiting a variety of behaviors, we find that (a) spatial components and temporal correlations (measured over timescales of tens of minutes) are consistent across different sessions in the same mouse, (b) the frontal areas of cortex are consistently useful in decoding the direction of licks in a spatial discrimination task, and (c) the parietal areas of cortex are useful in decoding the movements of the paws during the same task. We begin below by describing the model, and then describe applications to a number of datasets.

## Results

### Model

Here, we summarize the critical elements of the LocaNMF approach that enable the constrained spatiotemporal decomposition of WFCI videos; full details appear in the Methods section. Our proposed decomposition approach takes NMF as a conceptual starting point but enforces additional constraints to make the extracted components more reproducible and interpretable. Our overall goal is to decompose the denoised, hemodynamic-corrected, motion-corrected video $Y$ into $\hat{Y} = AC$, for two appropriately constrained matrices $A = \{\mathbf{a_k}\}$ and $C = \{\mathbf{c_k}\}$ (Fig 1). In more detail, we model

$$\hat{Y}(n, t) = \sum_k \mathbf{a}_k(n)\mathbf{c}_k(t), \tag{1}$$

i.e., we are expressing $\hat{Y}$ as the sum over products of spatial components $\mathbf{a}_k$ and temporal components $\mathbf{c}_k$. It is understood that each imaged pixel $n$ in WFCI data includes signals from a population of neurons visible at $n$, which may include significant contributions from neuropil activity [21]. Here, we assume that the term $\mathbf{a}_k(n)$ represents the density of calcium indicator at pixel $n$ governed by temporal component $k$, and is therefore constrained to be non-negative for each $n$ and $k$. $Y$, on the other hand, corresponds directly to the mean-adjusted fluorescence

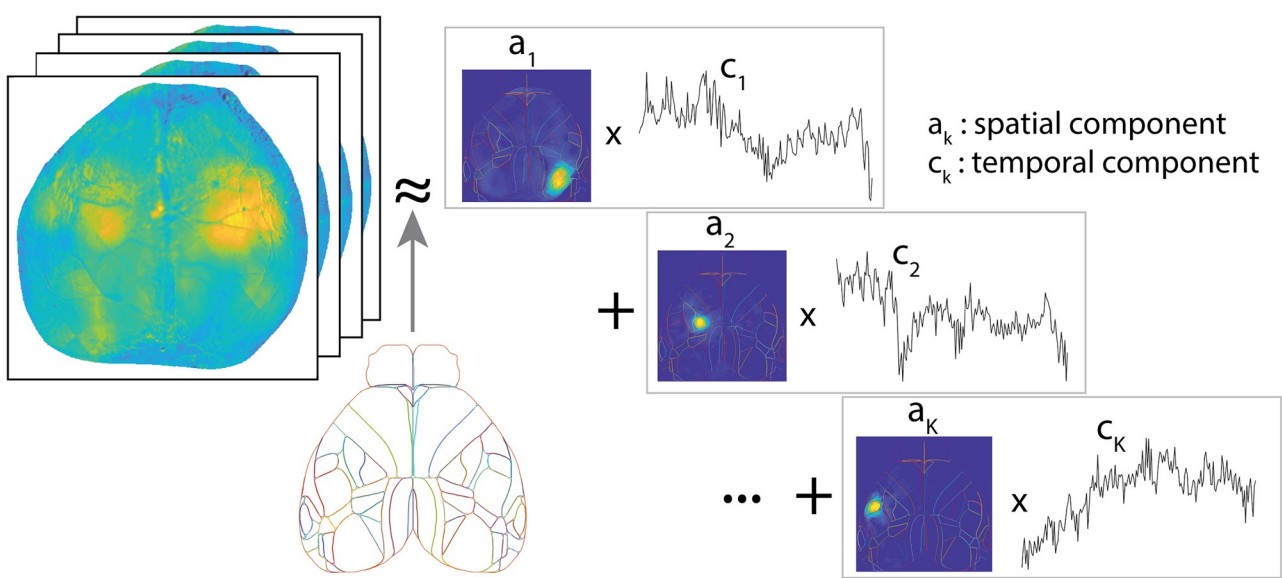

**Fig 1. Overview of LocaNMF: A decomposition of the WFCI video into spatial components *A* and temporal components *C*, with the spatial components soft-aligned to an atlas, here the Allen Institute Common Coordinate Framework (CCF) atlas.**

of every pixel ($\Delta F/F$), and as such may take negative values. Therefore, we do not constrain the temporal components *C* to be non-negative. Note that we are not making any assumption here about the cellular compartmental location of this calcium indicator density (e.g., somatic versus neuropil). For example, if the indicator is localized to the neuropil (or if the neuropil of the labeled neural population is superficial but the cell bodies are located more deeply), then a strong spatial component $\mathbf{a}_k$ in a given brain region may correspond to somatic activity in a different brain region.

The low-rank decomposition of *Y* into a non-negative spatial *A* matrix and a corresponding temporal *C* matrix falls under the general class of "semi-NMF" decomposition [23]. However, as detailed below, the components that we obtain using this decomposition are not typically interpretable; the spatial components can span the entire image due to the spatial correlations in the data. (Similar comments apply to principal components analysis or independent components analysis applied directly to *Y*). To extract more interpretable components as well as to compare activity across sessions and subjects, we would like to match each of them to a well-defined brain region. This corresponds to each component $\mathbf{a}_k$ being sparse, but in a very specific way, i.e., sparse outside the functional boundaries of a specific region. We use the Allen CCF brain atlas [24] to guide us while determining the initial location of the different brain regions, and constrain the spatial components to not stray too far from these region boundaries by including an appropriate penalization as we minimize the summed square residual of the factorization. Note that a different brain atlas could easily be swapped in here to replace the Allen CCF atlas, if desired.

To develop this decomposition, we first introduce some notation. We provide a summary of the notation in Table 1. We use a 2D projection of the Allen CCF map here, as in [8], which is partitioned into *J* disjoint regions $\Pi = \{\pi_1, \cdots, \pi_J\}$. Using LocaNMF, we identify *K* components. Specifically, each atlas region *j* gets $k_j$ components, possibly corresponding to different neural populations displaying coordinated activity, and $K: = \sum_j k_j$. Each component *k* maps to a single atlas region.

**Table 1. A summary of the notation for LocaNMF, with the corresponding matrix dimensions and descriptions.**

| Variable | Dimensions | Description |
|---|---|---|
| $N$ | $1 \times 1$ | Number of pixels in video |
| $T$ | $1 \times 1$ | Number of time points in video |
| $K_d$ | $1 \times 1$ | Rank of denoised video |
| $Y$ | $N \times T$ | Denoised video; $Y = UV$ |
| $U$ | $N \times K_d$ | Low-rank denoised spatial components |
| $V$ | $K_d \times T$ | Low-rank denoised temporal components |
| $L$ | $K_d \times K_d$ | Lower triangular matrix in the LQ decomposition of $V$; $V = LQ$ |
| $Q$ | $K_d \times T$ | Orthogonal matrix in the LQ decomposition of $V$; $V = LQ$ |
| $J$ | $1 \times 1$ | Number of regions predefined in the brain atlas. |
| $k_j$ | $1 \times 1$ | Number of LocaNMF components in $j^{th}$ region |
| $K$ | $1 \times 1$ | Total number of components found by LocaNMF; $K = \sum_{j=1}^{J} k_j$ |
| $A$ | $N \times K$ | LocaNMF spatial components |
| $C$ | $K \times T$ | LocaNMF temporal components |
| $\hat{Y}$ | $N \times T$ | LocaNMF decomposed video; $\hat{Y} = AC$ |
| $B$ | $K \times K_d$ | Multiplicative matrix in the decomposition of $C$; $C = BQ$ |
| $L_k$ | $1 \times 1$ | Localization constant for the $k^{th}$ component |
| $\Lambda$ | $K \times 1$ | Lagrangian parameters for the localization constraint in Eq 5. |

We solve the following optimization problem, where $Y \in \mathbb{R}^{N \times T}$:

$$min_{A,C} \quad \|Y - AC\|_F^2 \tag{2}$$

$$s.t. \quad A \geq 0, \quad \|\mathbf{a}_k\|_\infty = 1 \quad \forall k \in [1, K], \quad A \in \mathbb{R}^{N \times K} \tag{3}$$

$$C \in \mathbb{R}^{K \times T} \tag{4}$$

$$\sum_{n=1}^{N} |\mathbf{d}_k(n)\mathbf{a}_k(n)|^2 \leq L_k \quad \forall k \in [1, K], \tag{5}$$

where $N$ is the number of pixels and $T$ the number of frames in the video, $\|\mathbf{a}_k\|_\infty$ signifies $max_n |\mathbf{a}_k(n)|$, and Eq 5 signifies a $\mathcal{L}_2$ distance penalty term, where $\mathbf{d}_k(n)$ quantify the smallest euclidean distance from pixel $n$ to the atlas region corresponding to component $k$. $\{L_k\}$ are constants used to enforce localization.

## Application to simulated data

We begin by applying LocaNMF to decompose simple simulated data (Fig 2). We simulate each region $k$ to be modulated with a Gaussian spatial field centered at the region's spatial median, with a width proportional to the size of the region. The temporal components $C_{real}$ for the $K$ regions were simulated to be sums of sinusoids with additional Gaussian noise. Full details about the simulations are included in the Methods.

We ran the LocaNMF algorithm with localization threshold 70% (i.e., at least 70% of the mass of each recovered spatial component was forced to live on the corresponding Allen brain region; see Methods for details), and recovered the spatial and temporal components as shown in Fig 2. We also ran SVD for comparison, and aligned the recovered and true components

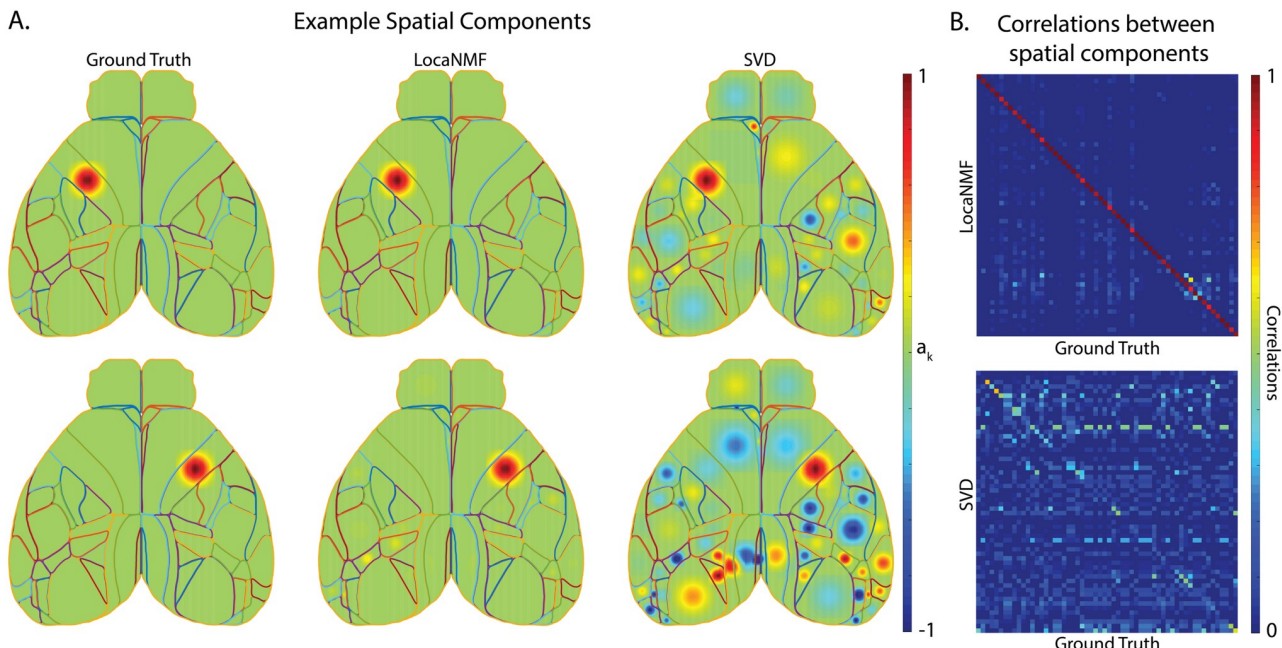

**Fig 2. LocaNMF can accurately recover the spatial and temporal components in simulated WFCI data.** (A) Left column: two example ground truth spatial components; Middle and Right columns: the corresponding spatial components as recovered by (Middle column) LocaNMF; (Right column) SVD. (B) Correlation between ground truth spatial components and those recovered by (Top) LocaNMF; (Bottom) SVD.

(by finding a matching that approximately maximized the $R^2$ between the real $A$ matrix and the recovered $A$ matrix). While LocaNMF recovered $A$ and $C$ accurately, SVD did not; there is a poor correspondence between the true $A$ and the $A$ recovered by SVD. Similar results held for vanilla NMF (here, vanilla semi-NMF; i.e., semi-NMF with no localization constraints); results are shown in S1 Fig.

## Application to experimental data

Next we applied LocaNMF to two real WFCI datasets. Data type (1) consisted of WFCI videos of size [540 × 640 × $T$], with $T$ ranging from 88, 653 to 129, 445 time points (sampling rate of 30Hz), from 10 mice expressing GCaMP6f in excitatory neurons. For each mouse, we analyzed movies from two separate experimental sessions recorded over different days. LocaNMF run on one GPU card (NVIDIA GTX 1080Ti) required a median of 29 minutes per session (on recordings of median length 1 hour) for this dataset. Data type (2) consisted of WFCI videos of size [512 × 512 × 5990] (sampling rate of 20Hz) from two sessions from one Thy1 transgenic mouse expressing jRGECO1a. See the Methods section for full experimental details. Unless mentioned explicitly, the analyses below are performed on data type (1).

We show an example LocaNMF decomposition for one trial with the mouse performing a visual discrimination task in this video, with localization threshold 80%. This shows the denoised brain activity for reference, and the modulation of the first two components LocaNMF extracted from each region, with different regions assigned different colors. We also display the rescaled residual as the normalized squared error between the denoised video and the LocaNMF reconstruction, as a useful visual diagnostic; in this case, we perceive no clear systematic signal that is being left behind by the LocaNMF decomposition.

In Fig 3 (left), we examine the top three components of the spatial maps of all regions across three different sessions from two different mice; we can see that the spatial maps are similar

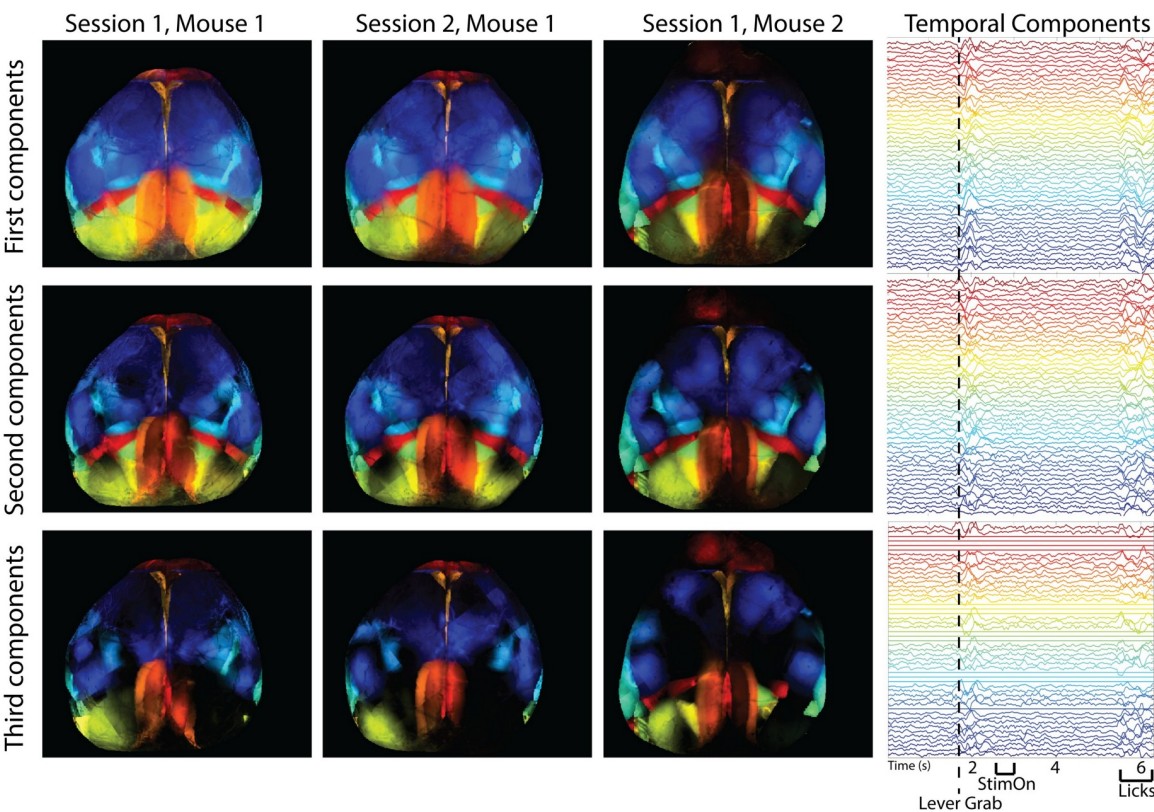

**Fig 3. Spatial and temporal maps of all regions in three different recording sessions from two different mice, as found with LocaNMF.** Note that LocaNMF outputs multiple components per atlas region. Left: the first, second and third component extracted from each region provided in each row, colored by region. Right: The trial-averaged temporal components for Session 1, Mouse 1 (aligned to lever grab), with the same color scheme as the spatial components. Link to a decomposed video of one trial here.

across sessions and mice (quantified across sessions in Fig 6, below). The trial-averaged temporal components on the right show modulations of a large number of components, time-locked to task-related behavioral events during the trial, consistent with recent results [8].

## Comparison with existing methods

**Comparison with region-of-interest analysis.** We implemented a decomposition that computes the mean denoised activity in each atlas brain region, otherwise known as a 'region-of-interest' (ROI) analysis with the atlas regions providing the ROIs. On a typical example session in dataset (1), this led to a mean $R^2 = 0.65$ (computed on the denoised data) as compared to the corresponding LocaNMF $R^2 = 0.99$; thus simply averaging within brain regions discards significant signal variance.

It is important to emphasize that the spatial components we obtain using LocaNMF are not simply confined to the atlas boundaries. To illustrate this point, we show two spatial components of one mouse in Fig 4A that extend past the corresponding atlas boundaries. Here, we show two spatial components anchored to the same atlas region that have very different spatial footprints $A_1$ and $A_2$, and moreover, have significantly different temporal components $C_1$ and $C_2$, respectively. The temporal components are also significantly different from $C_{ave}$, which is the temporal component that is obtained by simply averaging over the pixel-wise $\frac{\Delta F}{F}$ in that

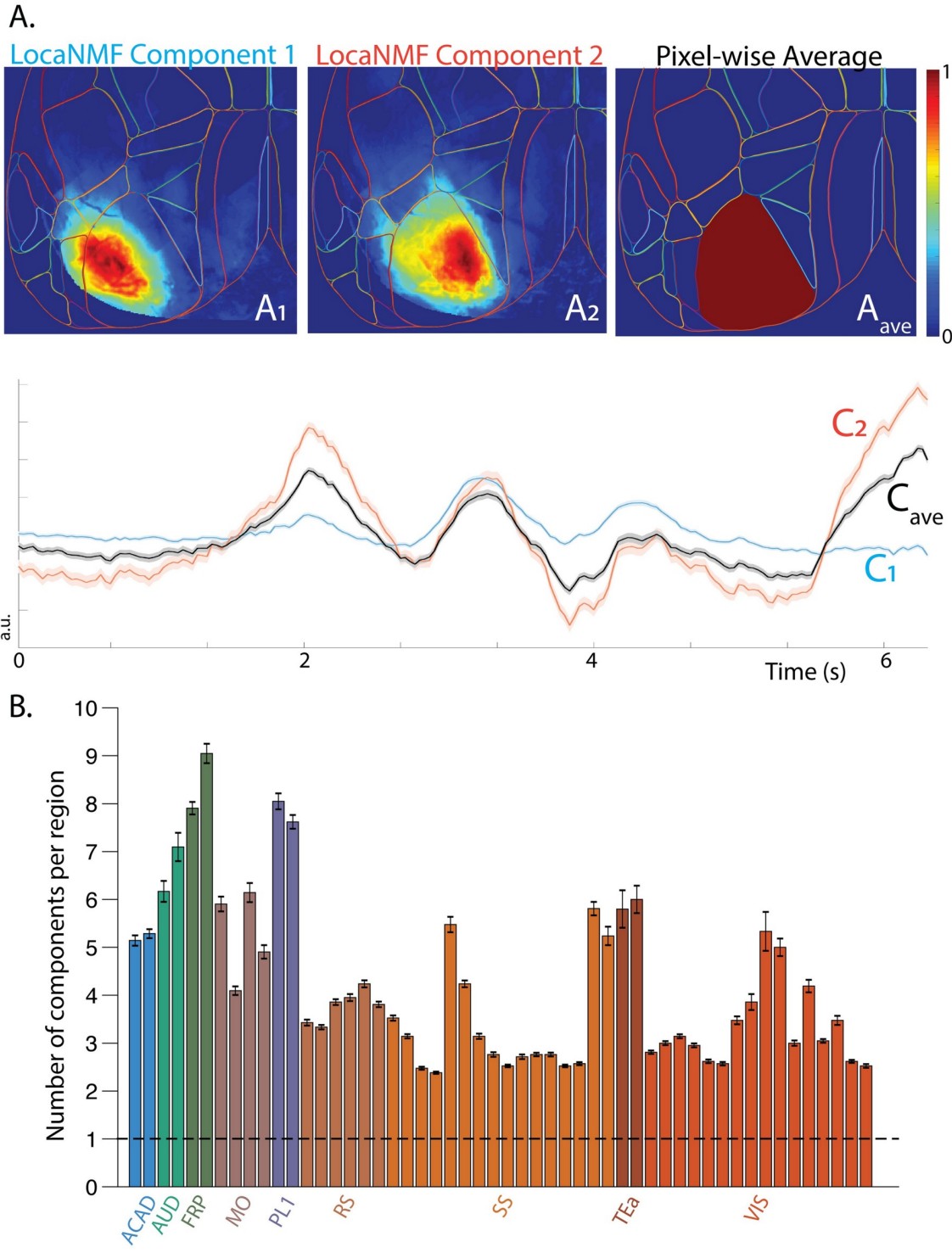

**Fig 4. Comparison with ROI analysis.** A. LocaNMF spatial components that are anchored to an Allen region show further specificity that may be lost if considering the average fluorescence in the Allen region as per an ROI analysis. B. The mean number of components recovered by LocaNMF. The bars are colored according to the cortical region they belong in, but note that there is one bar per subregion (ex. primary somatosensory cortex, right hand side upper limb). The dashed line at 1 signifies the number of components found with an ROI analysis.

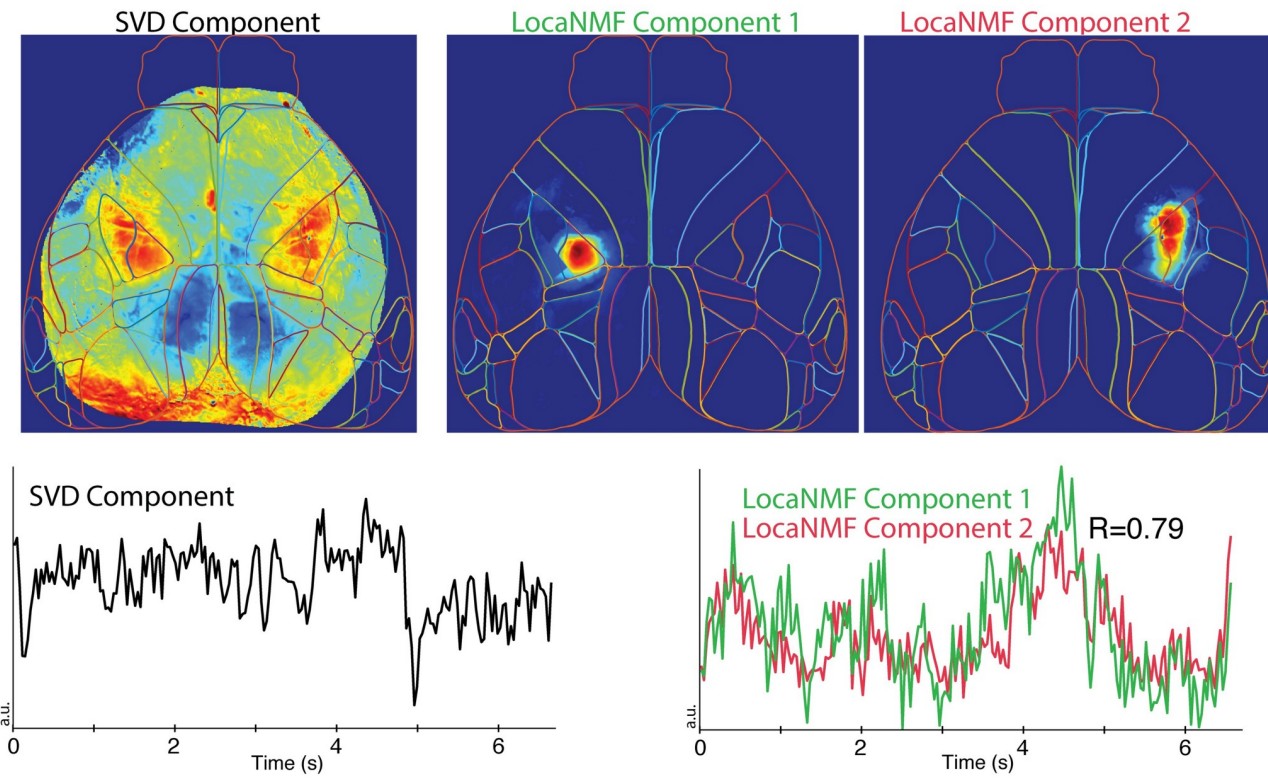

**Fig 5. LocaNMF can capture long range correlations that are difficult to analyze via SVD.** Top left: example de-localized spatial component recovered by SVD. This component places significant weight on multiple widely-separated brain regions. The corresponding temporal component is shown in the lower left panel. In the same dataset, two separate components are recovered by LocaNMF, capturing activity in each of the two distant brain regions activity (top middle and right panels). LocaNMF recovers two separate time courses here (lower right), allowing us to quantify the correlation between the regions (R = 0.79).

atlas region (left hand side primary visual cortex), illustrating that an ROI analysis discards significant spatiotemporal structure present in the data.

**Comparison with singular value decomposition.** Above we noted that simple SVD does a poor job of extracting the true spatial components from simulated data. In real data, we find that in many cases the SVD-based components are highly de-localized in space. In Fig 5, we see an example of an SVD component that represents activity across two distinct regions in the primary somatosensory cortex: the left hand side lower limb region and the right hand side upper limb region. In these cases LocaNMF simply outputs multiple components with correlated temporal activity, as shown in Fig 5. This allows us to quantify the correlations across regions (by computing correlations across the output temporal components), rather than just combining these activities into a single timecourse. See Figs 6 and 7 for additional examples of de-localized components output by SVD.

**Comparison with vanilla NMF.** LocaNMF can be understood as a middle ground between two extremes. If we enforce no localization, we obtain vanilla NMF with an atlas initialization. Alternatively, if we enforce full localization (i.e., force each spatial component $\mathbf{a}_k$ to reside entirely within a single atlas region), we obtain a solution in which NMF is performed independently on the signals contained in each individual atlas region. (Note that even in this case we typically obtain multiple signals from each atlas region, instead of simply averaging over all pixels in the region.) Across the 20 sessions in 10 mice in dataset (1), this fully-localized per-region NMF requires an average of 452 total components to reach our reconstruction

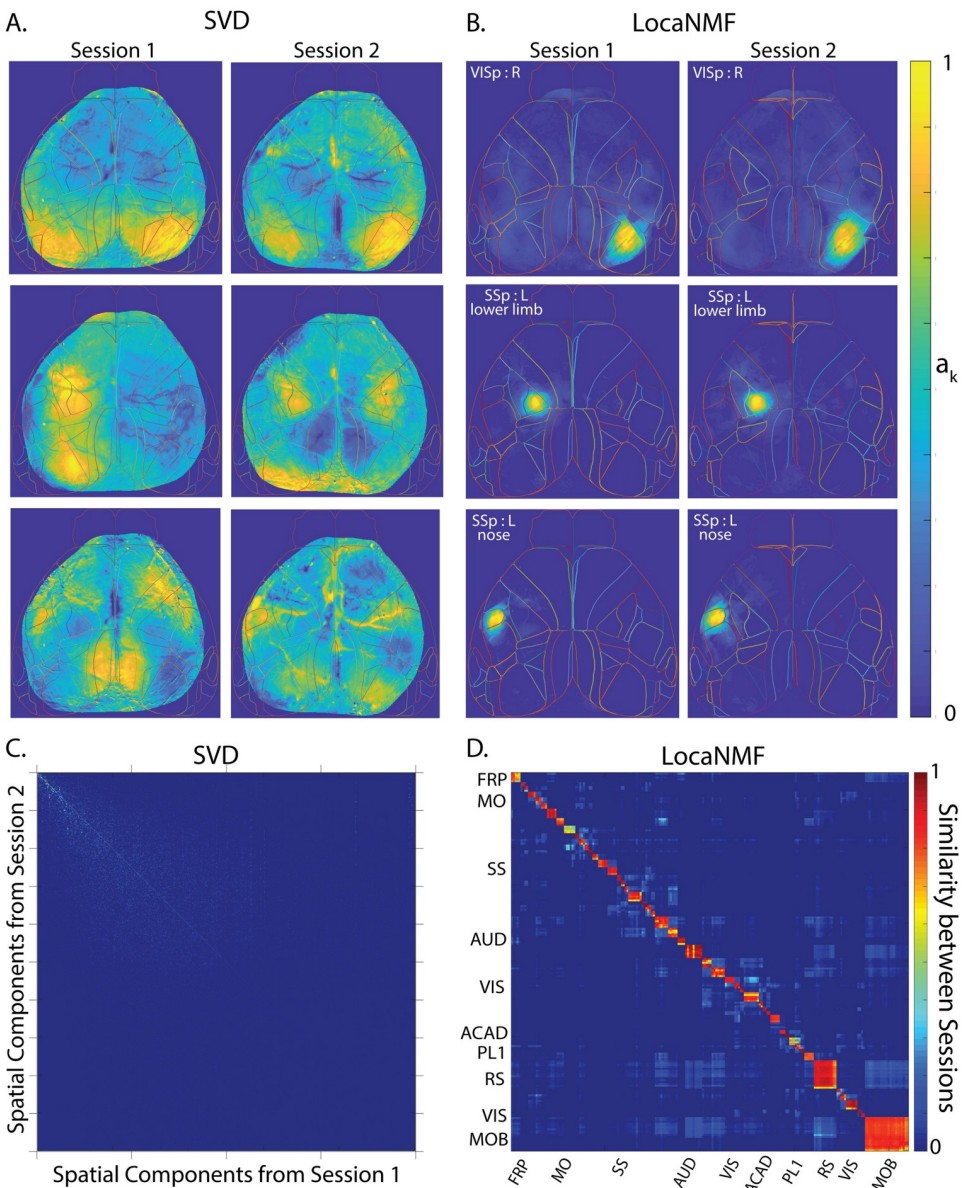

**Fig 6. LocaNMF extracts localized spatial components that are consistent across two recording sessions across different days (session length = 49 and 64 minutes; in each case the mouse was performing a visual discrimination task).** Example spatial components extracted from three different regions and two different sessions for one mouse expressing GCaMP6f, using A. SVD, and B. LocaNMF as in Algorithm 1. Note that LocaNMF components are much more strongly localized and reproducible across sessions. Cosine similarity of spatial components across two sessions in the same mouse using C. SVD after component matching using a greedy search, and D. LocaNMF. As in the simulations, note that LocaNMF components are much more consistent across sessions.

accuracy threshold ($R^2_{thr} = 0.99$) on denoised data, while vanilla NMF requires on average 188 components to capture the same proportion of variance. Meanwhile, LocaNMF with a localization threshold of 80% outputs an average of 205 components (with the same accuracy threshold); thus enforcing locality on the LocaNMF decomposition does not lead to an overinflation of the number of components required to capture most of the variance in the data.

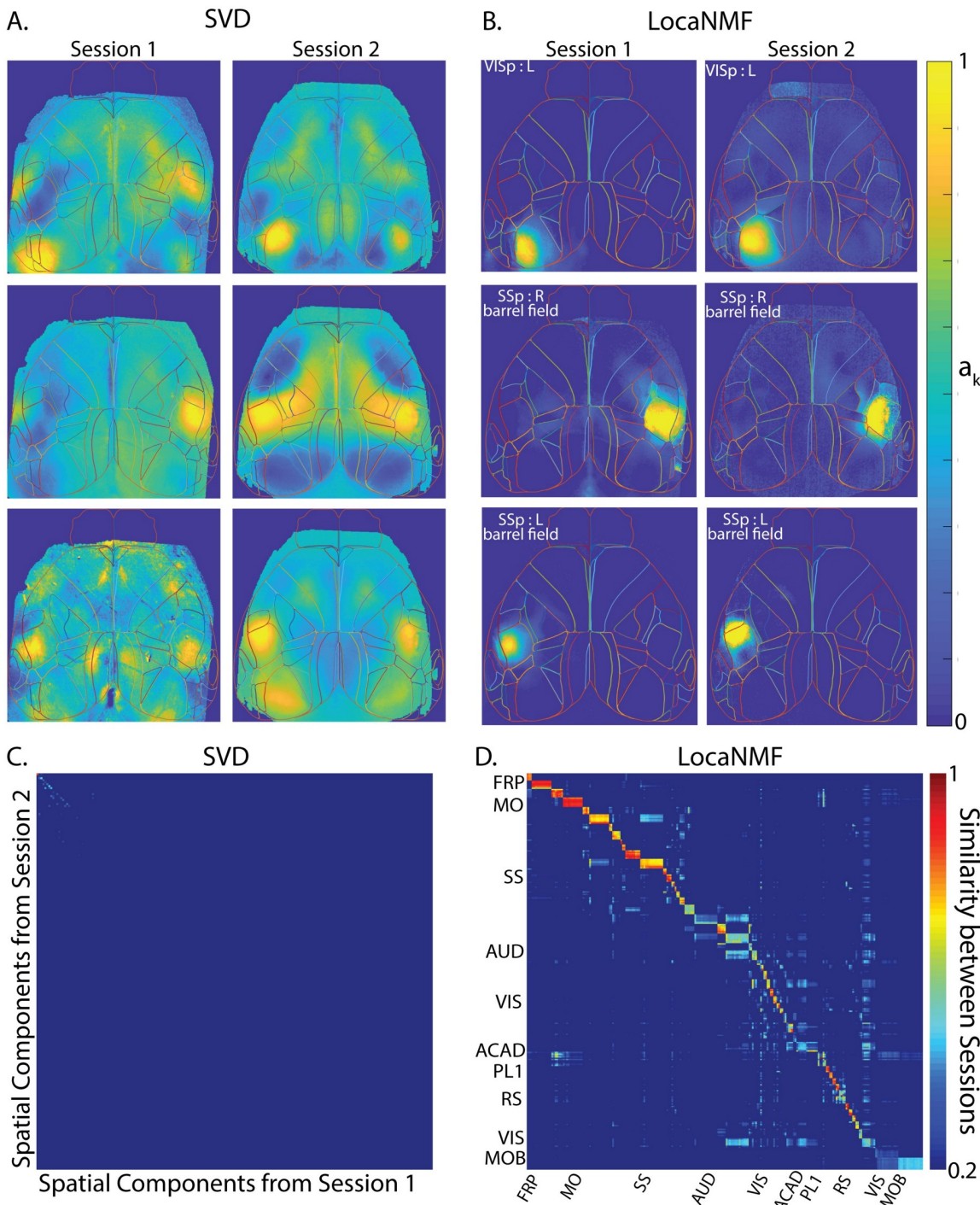

**Fig 7. LocaNMF applied to data from a mouse expressing jRGECO1a, with sessions of length 5 minutes.** A-D. Legend and conclusions similar to Fig 6A–6D.

The results for all figures showing SVD are also shown using vanilla NMF with random initialization in the Supplementary Information (see S1, S2, S4 and S5 Figs). Note that this method is initialization dependent and thus leads to different results even when run multiple times on the same dataset (see S3 Fig). While vanilla NMF with an atlas initialization addresses

this issue, it still leads to non-localized components which are not comparable across sessions (see S6 Fig).

## LocaNMF outputs localized spatial maps that are consistent across experimental sessions

When recording two different sessions over different days in the same mouse while the mouse is performing the same behavior, it is natural to expect to recover similar spatial maps. To examine this hypothesis, we analyzed the decompositions of two different recording sessions in the same mouse (Fig 6); we then repeated this analysis using a different mouse from dataset (2) (Fig 7). In both datasets, LocaNMF outputs localized spatial maps that are consistent across experimental sessions, as shown in Figs 6C, 6D, 7C and 7D, whereas both SVD and vanilla NMF outputs components that are much less localized and much less consistent across sessions. The results for vanilla NMF with a random and atlas initialization are shown in S4, S5 and S6 Fig.

## Correlation maps of temporal components show consistencies across animals

Next, we wanted to examine the relationship between the temporal activity extracted from different mice. We apply LocaNMF to all 10 mice in dataset (1) and examine the similarities in correlation structure in the temporal activity across sessions and mice. Since LocaNMF provides us with multiple components per atlas region, and we wish to be agnostic about which components in one region are correlated with those in another region, we use Canonical Correlation Analysis (CCA) to summarize the correlations from components in one region to the components in another region. CCA maps for four sessions of 49–65 minutes each, from two different mice, are shown in Fig 8A. In all sessions, the mice were engaged in either a visual or an audio discrimination task. We see that we recover clear similarities across CCA maps computed at the timescale of tens of minutes in different recording sessions, and different animals. We find that CCA maps of different sessions in the same mouse tend to be more similar than are CCA maps of sessions across different mice, as quantified in Fig 8C.

## Event-driven temporal modulation of brain regions is consistent across mice and is time-locked to key behavioral markers

How are the components extracted by LocaNMF related to behaviorally relevant signals? To examine this question, we begin by examining the trial-averaged components extracted from each region (Fig 9A). We see significant lateralized modulation of the primary visual cortex following the onset of visual stimulation (see top row of Fig 9A for right side). We also see a significant bilateral modulation of the primary somatosensory cortex (upper limb area) time-locked to lever grab behavior (bottom row).

Next, we take the trial-averaged response of the LocaNMF components of each functional region while the mouse is licking the spout in the Left vs Right direction, and form a [*Direction* × *Components* × *Time*] tensor. We wanted to assess the dependence of the different regions' activity on the lick direction, and to quantify the consistency of this dependence across sessions. Demixed Principal Component Analysis [24] is a method designed to separate out the variance in the data related to trial type (e.g., lick direction) vs. variance related to other aspects of the trial such as time from lick event. We show the top demixed principal components of the trial-averaged response of the right hand side primary somatosensory area, mouth

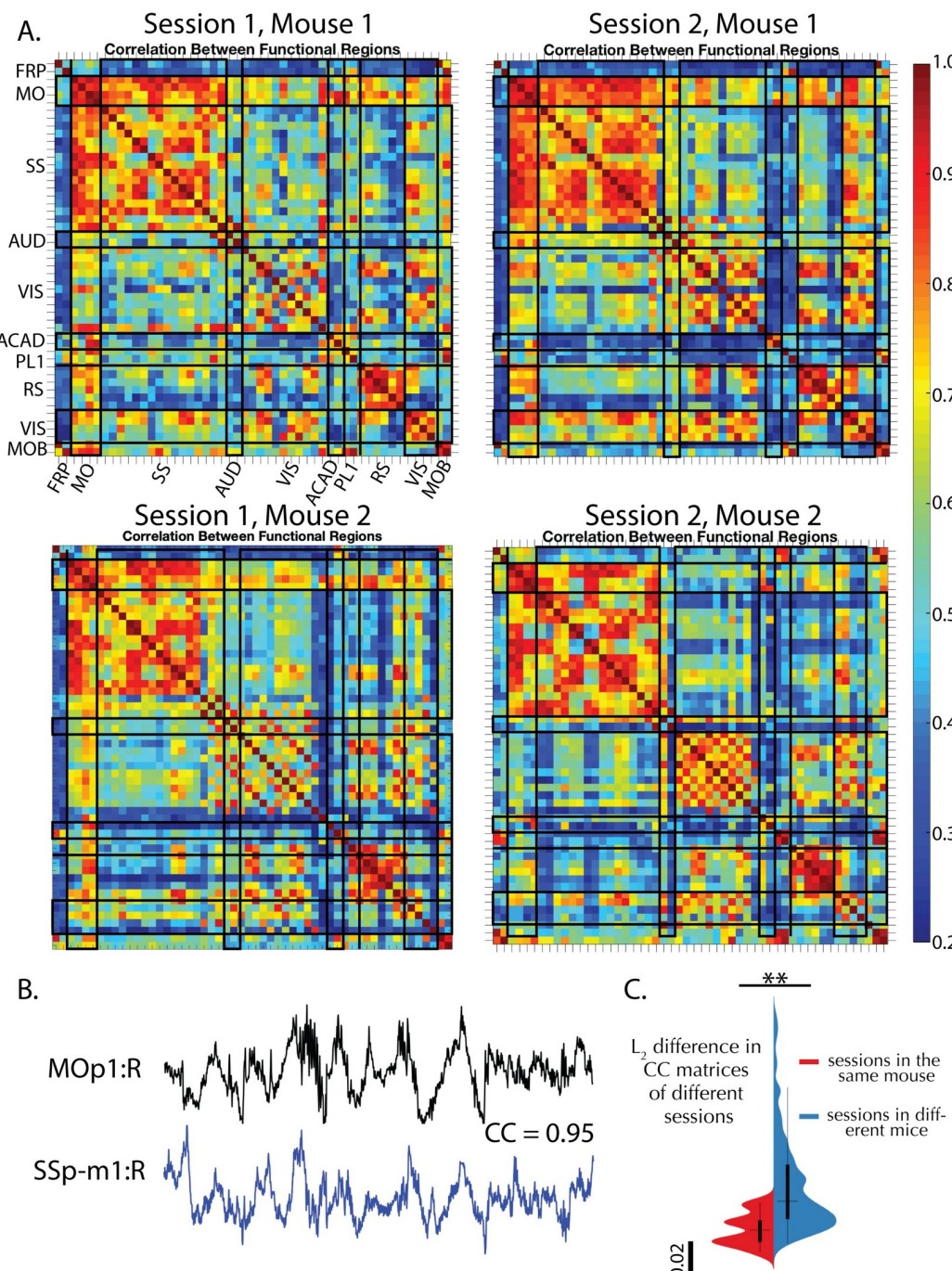

**Fig 8. Correlation maps of temporal components extracted by LocaNMF show consistencies acrosssessions and animals.** A. Top canonical correlation coefficient between the temporal components of any two regions, shown for four different sessions of 49 to 64 minutes each, recorded across two mice. B. Example traces of two highly correlated regions. C. Violin plot of mean squared difference between the correlation maps of the 20 different sessions across 10 mice; on average, within-mice differences are smaller than across-mice differences (One-tailed t-test $p = 0.0025$).

region (SSp-m1:R), and the right hand side of the secondary motor cortex (MOs1:R), of one mouse during two different sessions (Fig 9B). These can be interpreted as 1$D$ latent variables for the two lick directions, here capturing 87% ± 4% of the variance in the trial-averaged components. We see that these latents start modulating before lick onset, and continue modulating

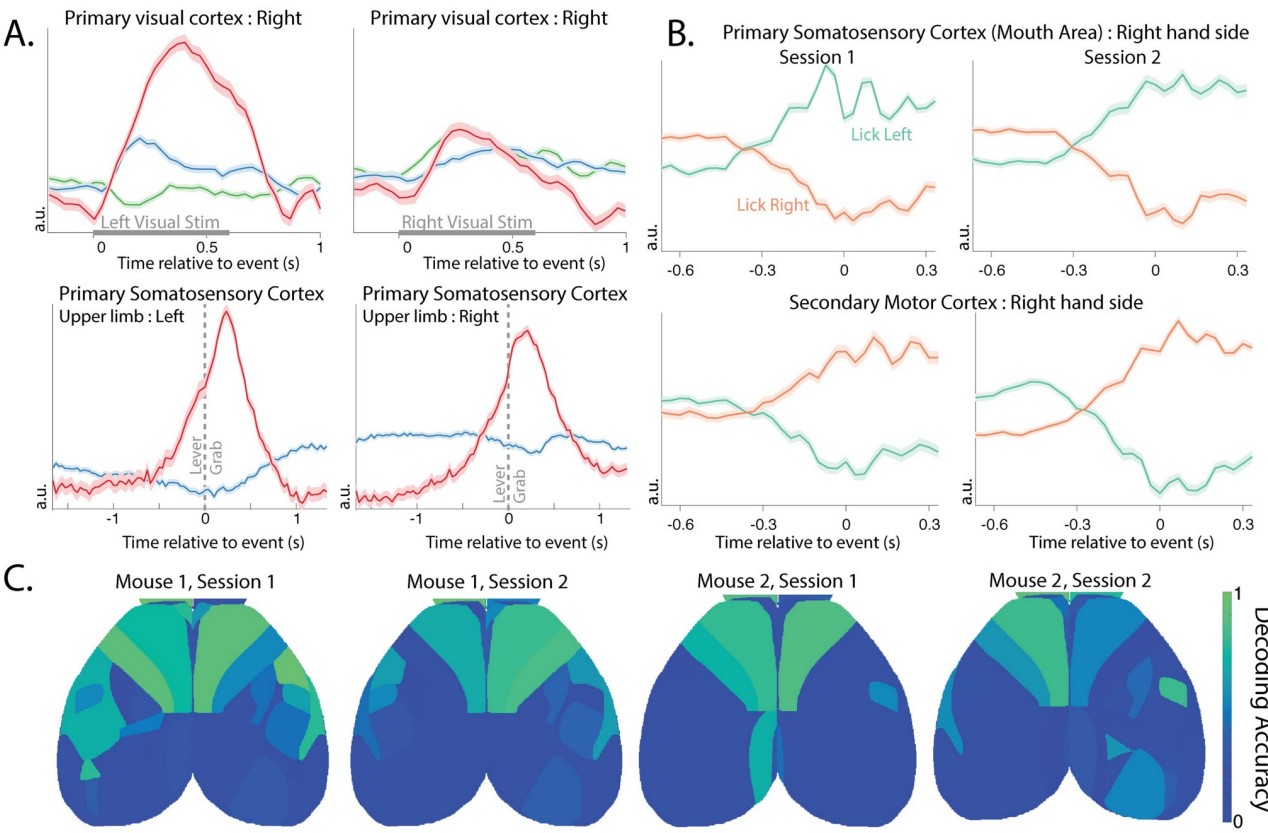

**Fig 9. Brain areas show consistencies in their activity around task-related behavior, and in their ability to decode direction of licking activity.** A. The LocaNMF components of the trial-averaged activity of the right hand side primary visual cortex (VISp) under left and right visual stimulus, and of the primary somatosensory area, upper limb area (SSp-ul), left and right hand sides, before and after the lever grab. Each color indicates a different component in the same region. Standard error of the mean is shaded. B. The top demixed Principal Component of the trial-averaged activity of the right hand side primary somatosensory area, mouth (SSp-m1:R) and right hand side secondary motor cortex (MOs1:R) before and after the onset of a lick to the left or right spout (onset at time 0). Standard error of the mean is shaded. The activity around licking left or right in both regions is consistent across the two sessions. C. Decoding accuracy on held-out data for the direction of lick (Left vs. Right spout) using only components in a shaded brain region. A logistic decoder was used on the time courses on data from 0.67s before and 0.33s after the event (lick left or lick right).

well past lick onset. Moreover, we see that the latents in these two areas modulate consistently across different sessions before and after a lick.

Finally, we use the activity of different brain regions to decode the direction of individual lick movements, i.e. the left (lickL) or right (lickR) direction on each instance of the lick movement. The input to the decoder on each lick instance consists of all of the temporal components from a given brain region, from 0.67s before each lick, up to lick onset (corresponding to 21 timepoints per temporal component). We build an $\mathcal{L}_2$ regularized logistic decoder based on this input to decode the direction of each lick (using 5-fold cross-validation to estimate the regularization hyperparameters). For data from held-out lick instances, we test the ability of each region's components to decode the lick direction (Fig 9C); we see that the frontal regions contain significant information that can be used to decode the lick direction.

## Decoding of behavioral components quantifies the informativeness of signals from different brain regions

Finally, we examine how the activity of different brain regions is related to continuous behavioral variables, rather than the binary behavioral features (i.e., lick left or right), addressed in

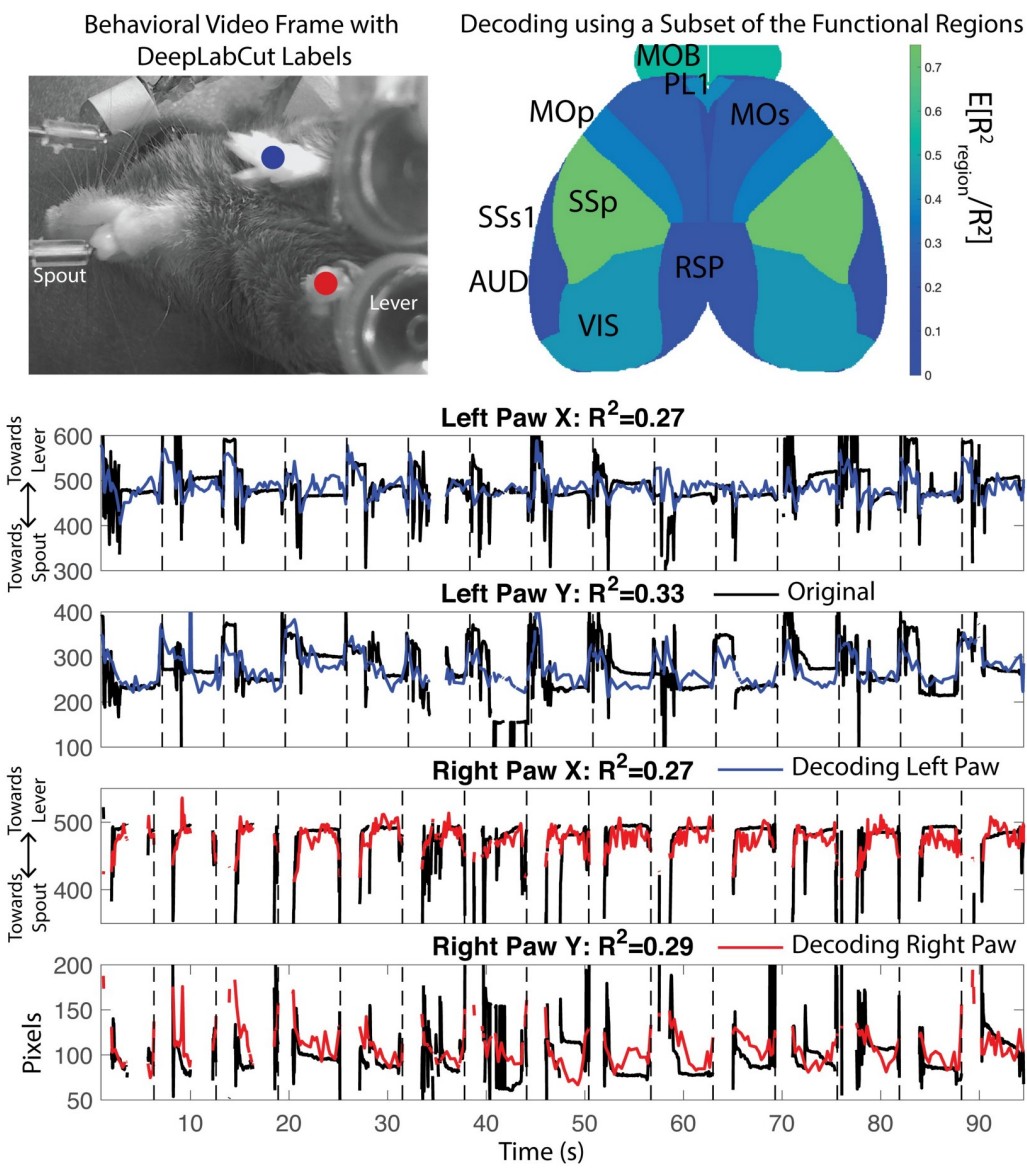

**Fig 10. Decoding paw position from WFCI signals.** Top Left: One frame of the DeepLabCut output, with decoded positions of left and right paws in blue and red. Top right: Relative decoding accuracy when the decoder was restricted to use signals from just one brain region, as a fraction of the $R^2$ using all signals from all brain regions. Area acronyms are provided in Table 2. Bottom: Decoding of DLC components using data from all brain regions for one mouse. Link to corresponding real-time videos for a few trials here, with DLC labels in black, and decoded paw location in blue and red for left and right paw respectively.

the preceding section. We tracked the position of each paw using DeepLabCut (DLC) [26] applied to video monitoring of the mouse during the behavior; an example frame is shown in Fig 10. We decoded the position of these markers using the temporal components extracted by LocaNMF (Fig 10 Bottom). (See Methods for full decoder details.) We found (a) that LocaNMF components are better at decoding paw locations than ROI components (mean $R^2 = 0.29$ with LocaNMF vs. 0.22 with ROI), and (b) that temporal signals extracted from the primary somatosensory cortex, the olfactory bulb, or the visual cortex lead to the highest decoding accuracy (Fig 10, top right). The primary somatosensory cortex may be receiving

proprioceptive inputs resulting from the movements of the paws, and the olfactory bulb is known to encode movements of the nose which may be correlated with the movements of the paws.

## Discussion

Widefield calcium recordings provide a window onto large scale neural activity across the dorsal cortex. Here, we introduce LocaNMF, a tool to efficiently and automatically decompose this data into the activity of different brain regions. LocaNMF outputs reproducible signals and enhances the interpretability of various downstream analyses. After having decomposed the activity into components assigned to various brain regions, this activity can be directly compared across exprimental preparations. For example, we build correlation maps that can be compared across different sessions and mice. Recently, several studies have shown the utility of having a fine-grained gauge of behavior alongside that of WFCI activity [8, 14]. We highlight that in order to have a more complete understanding of *how* the cortical activity may be leading to different behaviors, we first need an interpretable low dimensional space common to different animals in which the cortical activity may be represented.

Although we used the Allen atlas to localize and analyze the WFCI activity in this paper, LocaNMF is amenable to any atlas that partitions the field of view into distinct regions. As better structural delineations of the brain regions emerge, the anatomical map for an average mouse may be refined. In fact, it is possible to test different atlases using the generalizability of the resulting LocaNMF decomposition on different trials as a metric. As potential future work, LocaNMF could also be adapted to refine the atlas directly by optimizing the atlas-defined region boundaries to more accurately fit functional regions.

Analyses using other imaging modalities, particularly fMRI, have also faced the issue of needing to choose between interpretability (for example, as provided by more conventional atlas-based methods) and efficient unsupervised matrix decomposition (for example, as in PCA, independent component analysis, NMF, etc.) [27]. Typically, diffusion tensor tractography [28] or MRI [29, 30] can be used for building an anatomical atlas, and seed-based methods are used for obtaining correlations in fMRI data. In all these methods, a registration step is first performed on structural data (typically, MRI), thus providing data that is well aligned across subjects. More recently, graph theoretic measures as well as other techniques for characterizing the functional connections between different anatomical regions have become increasingly popular in fMRI [31–33]; these first perform a parcellation of the across-subject data into regions of interest (ROIs), then average the signals in each ROI before pursuing downstream analyses. Parcellations combining anatomical and functional data have also been pursued [34].

We view LocaNMF as complementary to these methods; here we perform an *atlas-based* yet data-driven matrix decomposition; importantly, instead of simple averaging of signals within ROIs we attempt to extract multiple overlapping signals from each brain region, possibly reflecting the contributions of multiple populations of neurons in each region. One very related study is [35], where the authors perform NMF on fMRI data, and introduce group sparsity and spatial smoothness penalties to constrain the decomposition. LocaNMF differs in the introduction of an atlas to localize the components; this directly enables across-subject comparisons and assigns region labels to the components (while still allowing the spatial footprints of the extracted components to shift slightly from brain to brain), which can be helpful for downstream analyses. Furthermore, recent studies have shown that the spatial and temporal activity recorded from WFCI and fMRI during spontaneous activity show considerable similarities [3, 36]. Given these conceptual similarities, we believe there are opportunities to adapt the methods we introduced here to fMRI or other three-dimensional (3D) functional

imaging modalities [37, 38], while using a 3D atlas of brain regions to aid in localization of the extracted demixed components. We hope to pursue these directions in future work.

## Methods

### Experimental details

**Data type (1).** Detailed experimental details are provided in [8]; we briefly summarize the experimental procedures below.

Ten mice were imaged using a custom-built widefield macroscope. The mice were transgenic, expressing the Ca2+ indicator GCaMP6f in excitatory neurons. Fluorescence in all mice was measured through the cleared, intact skull. The mice were trained on a delayed two-alternative forced choice (2AFC) spatial discrimination task. Mice initiated trials by making contact with their forepaws to either of two levers that were moved to an accessible position via two servo motors. After one second of holding the handle, sensory stimuli were presented for 600 ms. Sensory stimuli consisted of either a sequence of auditory clicks, or repeated presentation of a visual moving bar (3 repetitions, 200 ms each). For both sensory modalities, stimuli were positioned either to the left or the right of the animal. After the end of the 600 ms period, the sensory stimulus was terminated and animals experienced a 500 ms delay with no stimulus, followed by a second 600 ms period containing the same sensory stimuli as in the first period. After the second stimulus period, a 1000 ms delay was imposed, after which servo motors moved two lick spouts into close proximity of the animal's mouth. Licks to the spout corresponding to the stimulus presentation side were rewarded with a water reward. After one spout was contacted, the opposite spout was moved out of reach to force the animal to commit to its initial decision. Each animal was trained exclusively on a single modality (5 vision, 5 auditory).

Widefield imaging was done using an inverted tandem-lens macroscope (Grinvald et al., 1991) in combination with an sCMOS camera (Edge 5.5, PCO) running at 60 fps. The top lens had a focal length of 105 mm (DC-Nikkor, Nikon) and the bottom lens 85 mm (85M-S, Rokinon), resulting in a magnification of 1.24x. The total field of view was 12.4 x 10.5 mm and the spatial resolution was ∼20um/pixel. To capture GCaMP fluorescence, a 500 nm long-pass filter was placed in front of the camera. Excitation light was coupled in using a 495 nm long-pass dichroic mirror, placed between the two macro lenses. The excitation light was generated by a collimated blue LED (470 nm, M470L3, Thorlabs) and a collimated violet LED (405 nm, M405L3, Thorlabs) that were coupled into the same excitation path using a dichroic mirror (#87-063, Edmund optics). From frame to frame, we alternated between the two LEDs, resulting in one set of frames with blue and the other with violet excitation at 30 fps each. Excitation of GCaMP at 405 nm results in non-calcium dependent fluorescence (Lerner et al., 2015), we could therefore isolate the true calcium-dependent signal as detailed below.

Motion correction was carried out per trial using a rigid-body image registration method implemented in the frequency domain, with a given session's first trial as the reference image [39]. Denoising was performed separately on the hemodynamic and the GCaMP channels. The denoising step outputs a low-rank decomposition of $Y_{\text{raw}} = UV + E$ represented as an $N \times T$ matrix; here $UV$ is a low-rank representation of the signal in $Y_{\text{raw}}$ and $E$ represents the noise that is discarded. The output matrices $U$ and $V$ are much smaller than the raw data $Y_{\text{raw}}$, leading to compression rates above 95%, with minimal loss of visible signal. We use an established regression-based hemodynamic correction method [4, 8, 40], with an efficient implementation that takes advantage of the low-rank structure of the denoised signals. In brief, the hemodynamic correction method consists of low pass filtering a hemodynamic channel $Y_h$ (405nm illumination), then rescaling and subtracting this signal from the GCaMP channel $Y_g$ (473nm

illumination), in order to isolate a purely calcium dependent signal. We utilize the low-rank structure of the denoised data in order to perform the hemodynamic correction efficiently, i.e., we perform the low-rank decomposition separately for each channel, and then perform hemo-dynamic correction using the low rank matrices. Specifically, we obtain $Y_h = U_h V_h + E_h$ and $Y_g = U_g V_g + E_g$. We low pass filter $V_h$ ($2^{nd}$ order Butterworth filter with cutoff frequency 15Hz) to get $V_h^{lpf}$, and estimate parameters $b_i$ and $t_i$ for each pixel $i$ such that $(U_g)_i V_g = b_i (U_h)_i V_h^{lpf} + t_i$ using linear regression. We now obtain our hemodynamic corrected GCaMP activity $Y$ as the residual of the regression, i.e. $Y = U_g V_g - B U_h V_h^{lpf} + T$, where $B$ is a diagonal matrix with the terms $b_i$'s in the diagonal, and $T$ is a vector made by stacking the terms $t_i$. In fact, we keep the low rank decomposition of $Y$ as $UV$, with $U = [U_g - B U_h \ T]$ and $V = [V_g; \ V_h^{lpf}; \ \mathbf{1}]$, where $U \in \mathbb{R}^{N \times K_d}$, $V \in \mathbb{R}^{K_d \times T}$. We then convert this value into a mean-adjusted fluorescence value of every pixel ($\Delta F/F$).

**Data type (2).**   For this dataset we imaged adult Thy1-jRGECO1a mice (line GP8.20, purchased from Jackson Labs) [41]. In preparation for widefield imaging, a thinned-skull craniotomy was performed over the cortex, in which the mouse was anesthetized with isoflurane, had its skull thinned, and was implanted with an acrylic headpiece for restraint. The mouse underwent a two-day post operative recovery period and were habituated to head-fixation and wheel running for two days. To perform the imaging, we head-fixed the mouse on a circular wheel with rungs. The mouse was free to run for approximately 5 minutes at a time, while an Andor Zyla sCMOS camera was used to capture widefield images 512x512 pixels in size, at 60 frames per second (fps), with an exposure time of 23.4 ms. To collect fluorescence data along with hemodynamic data, we used three LEDs which were strobed synchronously with frame acquisition, producing an effective frame rate of 20 fps. Two LEDs were strobed to capture hemodynamic fluctuations (green: 530nm with a 530/43 bandpass filter and red: 625nm), and a separate LED (lime: 565 nm with a 565/24 bandpass filter) was strobed to capture fluorescence from jRGECO1a. A 523/610 bandpass filter placed in the path of the camera lens to reject emission LED light. Once collected, images were processed to account for hemodynamic contamination of the neural signal. Red and green reflectance intensities were used as a proxy for hemodynamic contribution to the lime fluorescence channel. The differential path length factor (DPF) was estimated and applied to calculate the DF/F neural signal. We performed hemodynamic correction as in [18], and then performed the denoising by performing SVD and keeping the top 200 components. Note that this also outputs a low-rank decomposition $Y_{raw} = UV + E$. Although the resulting $Y = UV$ is an efficient decomposition of the data, it consists of delocalized, uninterpretable components, as shown in the Results section.

## Details of simulations

We use LocaNMF to decompose simulated data (Fig 2). We simulate each region $k$ to be modulated with a gaussian spatial field with centroid at the region's median, and a width proportional to the size of the region ($\sigma_k = 0.2 \sqrt{(d_k)}$, where $d_k$ is the number of pixels in region $k$). The spatial components are termed $A_{real}(k)$, and were 534x533 pixels in size. The temporal components for the $K$ regions in simulated datasets (1) and (2) were specified as the following.

$$C_{real}(k) \quad \sim \quad \mathcal{N}\left(\sum_{j=1}^{3} \alpha_{jk} \sin(\beta_{jk} t), \ 0.1\right) \quad \forall k \in [1, K] \tag{6}$$

$$\alpha_{jk} \quad \sim \quad U(-1.5, 1.5) \quad \forall j, k \tag{7}$$

$$\beta_{jk} \quad \in \quad \{\beta'_1, \ldots, \beta'_{10}\}, \quad \forall j, k \tag{8}$$

$$\beta'_i \quad \sim \quad U(0.5, 0.63) \quad \forall i \in [1, 10]. \tag{9}$$

Here, $U(a, b)$ denotes the continuous uniform distribution on the interval $(a, b)$. We simulated $10{,}000$ time points at a sampling rate of 30Hz, and specified the decomposition $U = A_{real}$, and $V = C_{real}$.

## Preprocessing: Motion correction, compression, denoising, hemodynamic correction, and alignment

We analyze two datasets in this paper; full experimental details are provided above. After motion correction, imaging videos are denoted as $Y_{\text{raw}}$, with size $N \times T$, where $N$ is the total number of pixels and $T$ the total number of frames. $NT$ may be rather large ($\geq 10^{10}$) in these applications; to compress and denoise $Y_{\text{raw}}$ as detailed above, we experimented with simple singular value decomposition (SVD) approaches as well as more sophisticated penalized matrix decomposition methods [20]. We found that the results of the LocaNMF method developed below did not depend strongly on the details of the denoising / compression method used in this preprocessing step.

As is well-known, to interpret WFCI signals properly it is necessary to apply a hemodynamic correction step, to separate activity-dependent from blood flow-dependent fluorescence changes [18, 42]. We applied hemodynamic correction to both datasets as detailed above. Finally, for both datasets, we rigidly aligned the data to a 2D projection of the Allen Common Coordinate Framework v3 (CCF) [40] as developed in [8], using four anatomical landmarks: the left, center, and right points where anterior cortex meets the olfactory bulbs and the medial point at the base of retrosplenial cortex. We denote the denoised, hemodynamic-corrected video as $Y$ (i.e., $Y = UV$ after appropriate alignment).

More information about the Allen CCF is provided below.

## Details of localized Non-Negative Matrix Factorization (LocaNMF)

Here, we provide the algorithmic details of the optimization involved in LocaNMF, as detailed in Eqs 2–5; provided here again for the reader's convenience.

$$min_{A,C} \quad \|Y - AC\|_F^2$$

$$s.t. \quad A \geq 0, \quad \|\mathbf{a}_k\|_\infty = 1 \quad \forall k \in [1, K], \quad A \in \mathbb{R}^{N \times K}$$

$$C \in \mathbb{R}^{K \times T}$$

$$\sum_{n=1}^{N} |\mathbf{d}_k(n)\mathbf{a}_k(n)|^2 \leq L_k \quad \forall k \in [1, K],$$

We denote $\mathbf{D} \in \mathbb{R}^{N \times K}$ as the distance matrix comprising the entries $\mathbf{d}_k(n)$. A summary of the notation for this section is provided in Table 1.

**Spatial and temporal updates.** Hierarchical Alternating Least Squares (HALS) is a popular block coordinate descent algorithm for NMF [23] that updates $A$ and $C$ in alternating

fashion, updating each component of the respective matrices at a time. It is straightforward to adapt HALS to the LocaNMF optimization problem defined above. We apply the following updates for the spatial components in $A$ (where we are utilizing the low-rank form of $Y = UV$):

$$\mathbf{a}_k \quad \leftarrow \quad \left[ \mathbf{a}_k + \frac{1}{\mathbf{c}_k^T \mathbf{c}_k} \left( (YC^T)_k - A(CC^T)_k - \lambda_k \mathbf{d}_k \right) \right]_+ \tag{10}$$

$$= \quad \left[ \mathbf{a}_k + \frac{1}{\mathbf{c}_k^T \mathbf{c}_k} \left( U(VC^T)_k - A(CC^T)_k - \lambda_k \mathbf{d}_k \right) \right]_+ \tag{11}$$

Here, $[x]_+ = \max\{0, x\}$, $k \in \{1, \ldots, K\}$, and $\lambda_k$ is a Lagrange multiplier introduced to enforce Eq 5; we will discuss how to set $\lambda_k$ below. We normalize the spatial components $\{\mathbf{a}_k\}$ after every spatial update, thus satisfying the constraint $\|\mathbf{a}_k\|_\infty = 1$ for each $k$ in Eq 3.

The corresponding updates of $C$ are a bit simpler:

$$\mathbf{c}_k \quad \leftarrow \quad \mathbf{c}_k + \frac{1}{\mathbf{a}_k^T \mathbf{a}_k} \left( A^T Y_k - (A^T A)_k C \right) \tag{12}$$

$$= \quad \mathbf{c}_k + \frac{1}{\mathbf{a}_k^T \mathbf{a}_k} \left( (A^T U)_k V - (A^T A)_k C \right). \tag{13}$$

We can simplify these further by noting that each temporal component $\hat{c}_k$ for a given solution $\hat{C}$ is contained in the span of $V \in \mathbb{R}^{K_d \times T}$. Using this knowledge, we can avoid constructing the full matrix $C \in \mathbb{R}^{K \times T}$, and instead use a smaller matrix $B \in \mathbb{R}^{K \times K_d}$ by representing each component within a $K_d$-dimensional temporal subspace spanned by the columns of $V$. Specifically, we can apply an LQ-decomposition to $V$, to obtain $V = LQ$ where $L \in \mathbb{R}^{K_d \times K_d}$ is a lower triangular matrix of mixing weights and $Q \in \mathbb{R}^{K_d \times T}$ is an orthonormal basis of the temporal subspace. If we decompose $C$ as $C = BQ$, it becomes possible to avoid ever using $Q$ in all computations performed during LocaNMF (as detailed below). Thus, we can safely decompose $V = LQ$, save $Q$ and use $L$ in all computations of LocaNMF to find $A$ and $B$, and finally reconstruct $C = BQ$ as the solution for the temporal components. In the case where $K_d \ll T$, this leads to significant savings in terms of both computation and memory.

**Hyperparameter selection.** To run the method described above, we need to determine two sets of hyperparameters. One set of hyperparameters consists of the number of components in each region $\mathbf{k} = (k_1, \cdots, k_J)$, which dictate the rank of each region. Each component $k$ maps to a single atlas region. $\phi \colon \{1, \cdots, K\} \mapsto \{\pi_1, \cdots, \pi_J\}$ (surjective $K \geq J$). The second set of hyperparameters consists of the Lagrangian weights for each component $\Lambda = (\lambda_1, \cdots, \lambda_K)$, chosen to be the minimum value such that the localization constraint in Eq 5 is satisfied. These two sets of hyperparameters intuitively specify (1) that the signal in each region is captured well, and (2) that all components are localized, respectively. These hyperparameters can be set based on two simple, interpretable goodness-of-fit criteria that users can set easily: (1) the variance explained across all pixels belonging to a particular atlas region, and (2) how much of a particular spatial component is contained within its region boundary. These can be boiled down to the following easily specified scalar thresholds.

1. $R_{thr}^2$: a minimum acceptable $R^2$ to ensure the neural signal for all pixels in an atlas region's boundary is adequately explained

2. $L_{thr}$: the percentage of a particular region's spatial component that is constrained to be inside the atlas region's boundary

The procedure consists of a nested grid search wherein a sequence of proposals $\mathbf{k}^{(0)}$, $\mathbf{k}^{(1)}$, ... are generated and for each $\mathbf{k}^{(n)}$ a corresponding sequence $\mathbf{\Lambda}^{(n,0)}$, $\mathbf{\Lambda}^{(n,1)}$, ... are proposed. We term $k_j$ the local-rank of region $j$. Intuitively, we wish to restrict the local-rank in each region as much as possible while still yielding a sufficiently well-fit model. Moreover, for each proposed $\mathbf{k}^{(n)}$, we wish to select the lowest values for $\mathbf{\Lambda}$, while still ensuring that each component is sufficiently localized. In order to achieve this, each layer of this nested search uses adaptive stopping criteria based on the following statistics for the $j^{th}$ region and $k^{th}$ component.

$$R^2(j) := 1 - \frac{1}{|\pi_j|} \sum_{n \in \pi_j} \frac{\|Y(n) - \hat{Y}(n)\|_2^2}{\|Y(n) - \bar{Y}(n)\|_2^2} = 1 - \frac{1}{|\pi_j|} \sum_{n \in \pi_j} \frac{\|U(n)L - A(n)B\|_2^2}{\|U(n)L - U(n)\bar{L}\|_2^2} \tag{14}$$

$$L(k) := \frac{\sum_{n \in \phi(k)} \mathbf{a}_k(n)^2}{\|\mathbf{a}_k\|_2^2} \tag{15}$$

Here, $Y(n)$ and $A(n)$ denote the value of these matrices at pixel $n$. Note that the right hand side term in Eq 14 is computationally less expensive, as detailed in the following subsection. The algorithm terminates as soon as a pair $(\mathbf{k}^{(n)}, \mathbf{\Lambda}^{(n,m)})$ yields a fit satisfying $R^2(j) \geq R_{thr}^2 \ \forall j$ and $L(k) \geq L_{thr} \ \forall k$.

*Details of the LQ decomposition of V.* We show here that we can perform *LQ* decomposition of *V* at the beginning of LocaNMF, proceed to learn *A*, *B* using LocaNMF as in Algorithm 1, and reconstruct $C = BQ$ at the end of LocaNMF, without changing the algorithm or the optimization function. The term *C* is traditionally used in (1) the spatial updates, (2) the temporal updates, and (3) computing the optimization function. Here, we address how we can replace *C* by *B* in each of these computations.

1. For the spatial updates in Eq 11, we need two quantities; namely (1) $U(VC^T)$ and (2) $A$ $(CC^T)$. We can use the decompositions $V = LQ$ and $C = BQ$ to the two quantities; (1) $U$ $(VC^T) = U(LQQ^TB^T) = U(LB^T)$ and (2) $A(CC^T) = A(BQQ^TB^T) = A(BB^T)$.

2. For the temporal update in Eq 13, using the LQ decomposition, we set $C = BQ = (A^TA)^{-1}$ $A^TULQ$; thus it suffices to update *B* to $(A^TA)^{-1}A^TUL$. The spatial and temporal updates are also detailed in Algorithms 3 and 4.

3. Finally, we need to compute the errors in Eq 14. We note that $\|Y(n) - \hat{Y}(n)\|_2^2 = \|U(n)V - A(n)C\|_2^2 = \|(U(n)L - A(n)B)Q\|_2^2 = \|U(n)L - A(n)B\|_2^2$. While computing $UV$ and $AC$ have a computational complexity of $\mathcal{O}(NK_dT)$ and $\mathcal{O}(NKT)$ respectively, this operation decreases the computational cost to $\mathcal{O}(NK_d^2)$ and $\mathcal{O}(NKK_d)$; for *T* large, this denotes a significant saving in both memory and time taken for the algorithm.

Thus, we do not need the term *Q* for the bulk of the computations involved in LocaNMF, making the algorithm considerably more efficient.

*Adaptive number of components per region.* We wish to restrict the local-rank in each region as much as possible while still yielding a sufficiently well-fit model. In order to do so, we gradually move from the most to least-constrained versions of our model and terminate as soon as the region-wise $R^2$ is uniformly high as determined by the threshold $R_{thr}^2$. Specifically, we iteratively fit a sequence of LocaNMF models. The search is initialized with $\mathbf{k}^{(0)} = \mathbf{1}_J k_{min}$ and after

each fit $\hat{\mathbf{Y}}^{(iter_K)} = \hat{\mathbf{A}}^{(iter_K)}\hat{\mathbf{C}}^{(n)}$ is obtained, set

$$
k_j^{(iter_K+1)} = \begin{cases} k_j^{(iter_K)} + 1 & \text{if } R^{2(iter_K)}(j) < R_{thr}^2 \\ k_j^{(iter_K)} & \text{otherwise} \end{cases}
$$

until $R^{2(iter_K)}(j) \geq R_{thr}^2 \ \forall j = 1, \cdots, J$.

*Adaptive* $\lambda$. For brain regions that have low levels of activity relative to their neighbors, or have a smaller field of view, it is possible that the activity of a large amplitude neighboring region is represented instead of the original region's activity. However, we do not want to cut off the spread of a component in an artificial manner at the region boundary. Thus, we impose the smallest regularization possible while still ensuring that each component is sufficiently localized. To do so, we will gradually move from the least constrained (small $\lambda$) to most constrained (large $\lambda$) model, terminating as soon as the minimum localization threshold is reached. The search is initialized with $\mathbf{\Lambda}^{(0)} = \mathbf{1}_K \lambda_{min}$ and after each fit $\hat{\mathbf{Y}}^{(iter_\lambda)} = \hat{\mathbf{A}}^{(iter_\lambda)}\hat{\mathbf{C}}^{(iter_\lambda)}$ is obtained, set

$$
\lambda_k^{(iter_\lambda+1)} = \begin{cases} \tau \lambda_k^{(iter_\lambda)} & \text{if } L^{(iter_\lambda)}(k) < L_{thr} \\ \lambda_k^{(iter_\lambda)} & \text{otherwise} \end{cases}
$$

until $L^{(iter_\lambda)}(k) \geq L_{thr} \ \forall k = 1, \cdots, K$. This requires a user-defined $\lambda$-step, $\tau = 1 + \epsilon$, where $\epsilon$ is generally a small positive number.

*Initialization*. Finally, for a fixed set of hyperparameters $\mathbf{\Lambda}$, $\mathbf{k}$ the model fit is still sensitive to initialization (since the problem is non-convex). Hence, in order to obtain reasonable results we must provide a data driven way to initialize all $K = \sum_{j=1}^J k_j$ components.

To initialize each iteration of the local-rank line search, the components for each region are set using the results of standard semi-NMF (sNMF) fits to their respective regions. To facilitate this process, a rank $k_{max}$ SVD is precomputed within each individual region and reused during each initialization phase. For a given initialization, denote the number of components in region $j$ as $k_j$. The initialization is the result of a rank $k_j$ sNMF fit to the rank $k_{max}$ SVD of each region. The components of these initializations are themselves initialized using the top $k_j$ temporal components of each within-region SVD. This is summarized in Algorithm 2.

*Computation on a GPU*. Most of the steps of LocaNMF involve large matrix operations which are well suited to parallelization using GPUs. While the original data may be very large, $U$ and $L$ are relatively much smaller, and often fit comfortably within GPU memory in cases where $Y$ does not. Consequently, implementations which take low rank structure into account may take full advantage of GPU-acceleration while avoiding repeated memory transfer bottlenecks. Specifically, after the LQ decomposition of $V$, we load $U$ and $L$ into GPU memory once and keep them there until the Algorithm 1 has terminated. This yields a solution $\hat{A}, \hat{B}$ which can transferred back to CPU in order to reconstruct $\hat{C} = \hat{B}Q$. We provide both CPU and GPU implementations of the algorithm in the code here.

*Decimation*. As in [43] and [20], we can decimate the data spatially and temporally in order to run the hyperparameter search, and then run Algorithm 1 once in order to obtain the LocaNMF decomposition $(A, C)$ on the full dataset. In this paper, we have not used this functionality due to speedups from using a GPU, but we can envision that it might be necessary for bigger datasets and / or limitations in computational resources.

*Computational cost.* The computational cost of LocaNMF is $\mathcal{O}(NK_dK)$ (assuming $N \geq K_d \geq K$), with the most time consuming steps being the spatial and temporal HALS updates. *maxiter*$_\lambda$ and *maxiter*$_K$ both provide a scaling factor to the above cost. Note that the computational scaling is also linear in $T$, but this just enters the cost twice, once during the LQ decomposition of $V$, and once more when reconstructing $C$ after the iterations; in practice, this constitutes a small fraction of the computational cost of LocaNMF.

**Algorithm 1**: Localized semi Nonnegative Matrix Factorization (LocaNMF)

```
Data: U, V, Π, D, R²ₜₕᵣ, Lₜₕᵣ, kₘᵢₙ, λₘᵢₙ, τ, maxitersK, maxitersλ,
maxitersHALS
Result: A, C
[L, Q] = LQ(V) # LQ decomposition of V
kⱼ ← kₘᵢₙ ∀j ∈ [1, J]
for iterK ← 1 to maxitersK do
  [A, B] ← Init-sNMF(U, L, Π, k, maxitersHALS)
  λₖ ← λₘᵢₙ ∀k ∈ [1, K]
  for iterλ ← 1 to maxitersλ do
    for iterHALS ← 1 to maxitersHALS do
      A ← HALSspatial(U, L, A, B, Λ, D)
      Normalize A
      B ← HALStemporal(U, L, A, B)
    end
    λₖ:Lₖ<Lₜₕᵣ ← τλₖ:Lₖ<Lₜₕᵣ
  end
  kⱼ:R²ⱼ<R²ₜₕᵣ ← kⱼ:R²ⱼ<R²ₜₕᵣ + 1
end
C = BQ
```

**Algorithm 2**: Initialization using semi Nonnegative Matrix Factorization (Init-sNMF)

```
Data: U, L, Π, k, maxitersHALS
Result: A, B
for j ← 1 to J do
  Uⱼ = U[πⱼ]
  Bⱼ = SVD(UⱼL, kⱼ); Aⱼ = 1[N×kⱼ]
  for iterHALS ← 1 to maxitersHALS do
    Aⱼ ← HALSspatial(Uⱼ, L, Aⱼ, Bⱼ)
    Normalize Aⱼ
    Cⱼ ← HALStemporal(Uⱼ, L, Aⱼ, Bⱼ)
  end
end
```

**Algorithm 3**: Localized spatial update of hierarchical alternating least squares (HALSspatial)

```
Data: U, L, A, B, D (defaults to 0[N×K]), Λ (defaults to 0[K])
Result: A
for k ← 1 to K do
```
$$\mathbf{a}_k \leftarrow \mathbf{a}_k + \left[\frac{1}{\mathbf{l}_k^T\mathbf{l}_k}\left(U(LB^T)_k - A(BB^T)_k - \lambda_k\mathbf{d}_k\right)\right]_+$$
```
end
```

**Algorithm 4**: Temporal update of hierarchical alternating least squares (HALStemporal)

```
Data: U, L, A, B
Result: B
for k ← 1 to K do
```
$$\mathbf{b}_k \leftarrow \mathbf{b}_k + \frac{1}{\mathbf{a}_k^T\mathbf{a}_k}\left((A^TU)_kL - (A^TA)_kB\right)$$
```
end
```

### Vanilla semi non-negative matrix factorization (vanilla NMF)

We use vanilla NMF with random initialization as a comparison to LocaNMF. When performing a comparison, we use the same number of components $K$ as found by LocaNMF. The algorithm is detailed in Algorithm 5.

**Algorithm 5**: vanilla semi-Nonnegative Matrix Factorization (vanilla NMF)

```
Data: U, V, K, maxiters_HALS
Result: A, C
A_k ~ B(N, 0.1) ∀k ∈ [1, K]# Bernoulli draws over pixels
C_k = E[(A_k ∘ U)V] ∀k ∈ [1, K]
for iter_HALS ← 1 maxiters_HALS do
  Normalize C
  A ← HALSspatial(U, V, A, C)
  C ← HALStemporal(U, V, A, C)
end
```

### Allen Common Coordinate Framework

The anatomical template of Allen CCF v3 as used in this paper is a shape average of 1675 mouse specimens from the Allen Mouse Brain Connectivity Atlas [44]. These were imaged using a customized serial two-photon tomography system. The maps were then verified using gene expression and histological reference data. For a detailed description, see the Technical White Paper here. The acronyms for the relevant components used in this study are provided in Table 2.

### Tracking parts in behavioral video

For the analysis involving the decoding of movement variables in the Results, we used DeepLabCut (DLC) [26] to obtain estimates of the position of the paws. We hand-labeled 144 frames as identified by K-means, with the locations of the right and left paws. We used standard package settings for obtaining the evaluations on all frames of one session.

For decoding the X and Y coordinate of each DLC tracked variable using inputs as the LocaNMF temporal components, we used an MSE loss function to train a one layer dense feedforward artificial neural network (64 nodes each, ReLu activations), with the last layer having as target output the relevant X or Y coordinate. We used 75% of the trials as training data (which is itself split into training and validation in order to implement early stopping), and we report the $R^2$ on the held out 25% of the trials.

**Table 2. Acronyms of the regions in the Allen atlas.**

| Acronym | Name |
| --- | --- |
| MOp | primary motor cortex |
| MOs | secondary motor cortex |
| SSp | primary somatosensory cortex |
| SSs1 | supplemental somatosensory cortex |
| AUD | auditory cortex |
| VIS | visual cortex |
| ACAd1 | anterior cingulate cortex (dorsal part) |
| PL1 | prelimbic cortex |
| RSP | retrosplenial cortex |

## Ethics statement

For data type (1), the Cold Spring Harbor Laboratory Animal Care and Use Committee approved all animal procedures and experiments. For data type (2), all procedures were reviewed and approved by the Columbia University Institutional Animal Care and Use Committee.

## Supporting information

**S1 Fig. Results of applying vanilla NMF to simulation data.** A-D. Legend and conclusions similar to Fig 2A–2D.
(TIF)

**S2 Fig. Results of applying vanilla NMF to uncover long-range correlations.** A-D. Legend and conclusions similar to Fig 5A–5D.
(TIF)

**S3 Fig. Cosine similarity across components after applying vanilla NMF with 10 different random initializations to an example session in dataset 1.** One of the randomly initialized vanilla NMF decomposition was chosen as the example decomposition, and each gray line shows the similarity between the components resulting from a different random initialization to this example decomposition, after component matching using a greedy search. The solid black line shows the mean similarity over initializations. The similarity across initializations is 1 for LocaNMF, shown here with a dashed black line.
(TIF)

**S4 Fig. Results of applying vanilla NMF to dataset 1.** A-D. Legend and conclusions similar to Fig 6A–6D.
(TIF)

**S5 Fig. Results of applying vanilla NMF to dataset 2.** A-D. Legend and conclusions similar to Fig 7A–7D.
(TIF)

**S6 Fig. Results of applying vanilla NMF with an atlas-based initialization to dataset 1.** There is more stability across sessions as compared to S4A Fig, but LocaNMF provides more stability still due to the localization constraint. A-D. Legend and conclusions similar to Fig 6A–6D.
(TIF)

**S7 Fig. The variance in the time courses of a regions' activity, where the activity in any one given region is the concatenation of the activity in all the subregions in that region.** The values are normalized by the maximum variance in a particular session. The values shown here are means over sessions, for all 20 sessions in 10 mice in dataset 1, with the standard error of the mean depicted around the mean.
(TIF)

## Acknowledgments

We thank Erdem Varol, Niko Kriegeskorte, and Catalin Mitelut for helpful conversations.

## Author Contributions

**Conceptualization:** Shreya Saxena, Simon Musall, Sharon H. Kim, Elizabeth M. C. Hillman, Anne Churchland, Liam Paninski.

**Data curation:** Simon Musall, Jozsef Meszaros, David N. Thibodeaux.

**Formal analysis:** Shreya Saxena, Ian Kinsella, Liam Paninski.

**Funding acquisition:** Shreya Saxena, Elizabeth M. C. Hillman, Anne Churchland, Liam Paninski.

**Investigation:** Shreya Saxena, Simon Musall, Sharon H. Kim, Jozsef Meszaros, David N. Thibodeaux, Carla Kim.

**Methodology:** Shreya Saxena, Liam Paninski.

**Project administration:** Liam Paninski.

**Resources:** Shreya Saxena, Ian Kinsella, Anne Churchland, Liam Paninski.

**Software:** Shreya Saxena, Ian Kinsella.

**Supervision:** John Cunningham, Elizabeth M. C. Hillman, Anne Churchland, Liam Paninski.

**Validation:** Shreya Saxena.

**Visualization:** Shreya Saxena, Simon Musall, Anne Churchland, Liam Paninski.

**Writing – original draft:** Shreya Saxena, Liam Paninski.

**Writing – review & editing:** Shreya Saxena, Ian Kinsella, Simon Musall, Jozsef Meszaros, John Cunningham, Elizabeth M. C. Hillman, Anne Churchland, Liam Paninski.

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
