## [Decision Letter · Decision Letter 0]

17 Dec 2019

Dear Dr Saxena,

Thank you very much for submitting your manuscript 'Localized semi-nonnegative matrix factorization (LocaNMF) of widefield calcium imaging data' for review by PLOS Computational Biology. Your manuscript has been fully evaluated by the PLOS Computational Biology editorial team and in this case also by independent peer reviewers. The reviewers appreciated the attention to an important problem, but raised some substantial concerns about the manuscript as it currently stands, particularly with regard to the adequacy of the model comparisons. While your manuscript cannot be accepted in its present form, we are willing to consider a revised version in which the issues raised by the reviewers have been adequately addressed. We cannot, of course, promise publication at that time.

Sincerely,

Samuel J. Gershman

Deputy Editor

PLOS Computational Biology

[LINK]

Reviewer's Responses to Questions

**Comments to the Authors:**

Reviewer #1: The paper presents a new Matrix Factorization Model for widefield calcium imaging data. The main contributions of this new model are:

- The Temporal components C are not constrained to be non-negative.

- The spatial components are initialized taking into account information from Allen CCF Brain Atlas. This information is also taken into account in the model, so as to encourage the spatial components to be coherent with them.

- The authors provide public CPU and GPU implementations of their methods in a complete suite for treating the entire widefield calcium process.

On the positive side, their approach incorporates knowledge (through the use of brain atlas) in the process and guides the solution towards biological meaning and stablity, through different runs. This can, indeed, be seen as a contribution to the most challenging shortcoming of Matrix Factorization methods: Their high dependence on the initialization of the components. Still, this doesn't fix the fact that the modeled computational problem is not solvable in polynomial time, which means the algorithms for solving the model may only be heuristic. Hence, the effectiveness of the technique can only be established through specific experiments. Indeed, the authors present interesting experiments that support their idea of constraining the location of the neurons to specific brain regions, which may help eliminate false positives.

The main drawback of the paper is in the comparison with only SVD. This is insufficient to support the experimental value of the proposed technique. Although the type of experiments performed are interesting and support the value of the work, I don't think that SVD methods, with which they compare, can be considered state-of-the-art. I think the paper has strong points, but should compare with techniques such as CNMF, Suite2P and HNCCorr. Furthermore, it would be interesting to see how the proposed method performs on the Neurofinder Dataset.

Reviewer #2: The main finding of this study is to propose a novel algorithm named ‘LocalNMF’ that decomposes widefield Ca imaging (WFCI) data into spatial and temporal components. The LocalNMF uses a brain atlas to initialize the estimated spatial components such that the spread of the spatial component is limited and localized in the different brain regions. The resulting components lead to a more interpretable decomposition of WFCI data. Its efficiency was validated with multiple experimental datasets related to binary behavioural features and continuous behavioural variables.

The authors did a superb job in designing the study and the experiments were conducted well. The data clearly support the conclusions of this study. The presentation of the results is good and the discussion is clear. Overall, this is a very interesting study that puts forward clear evidence for using NMF for analysing data revealed in neuroscience. I would like to recommend it for publication. However, the following comments may help to improve the writing and readability of the current version of the manuscript.

(1) Line 64: The main advancement of this study is stated as "a new approach to perform a localized, more interpretable decomposition of WFCI data. The proposed approach is a variation on classical NMF, termed localized semi-NMF (LocaNMF), that decomposes the widefield activity by (a) using existing brain atlases to initialize the estimated spatial components, and (b) limiting the spread of each spatial component in order to obtain localized components." Thus the authors emphasize that both (a) and (b) are important. If I understand it correctly, these points are related to the discussion on Sec. Comparison with vanilla NMF on line 200. My question here is about the results by using vanilla NMF, e.g., four Suppl. figures. Initially, I thought that these results are obtained by the vanilla NMF with the SAME brain atlas initialization. However, in the end, the Methods part at line 576, I found that "We use vanilla NMF with random initialization as a comparison to LocaNMF". Thus for comparison, there are some incomplete issues. For vanilla NMF, it can capture similar behaviors as in LocaNMF. Indeed, since the vanilla NMF or NMF, in general, is a method for detecting the localized feature. As shown in all of four SI figures, NMF can capture some similar behaviors as in locaNMF. For example, as in Fig SI 4, SSp : L barrel field is comparable for both NMF and locaNMF. Therefore, the localNMF employs a localization constraint (Eq. 5) to improve the robustness of the NMF. This is an advantage to overcome the non-robustness. I wonder if other tricks can also play the same role. There are many tricks to get rid of the local minimums during the convergence of NMF. Certainly, the current one, locaNMF, may be the most efficient and best one in this application scenario. However, to demonstrate this, one snapshot of the results is not enough, i.e., the results of four SI figures are outcomes of one single run of vanilla NMF. For the same dataset, suppose there are 10 runs of NMF, both vanilla NMF and locaNMF, the robustness of the results can be seen from all of the 10 runs, where each run has some random effect. I guess the reasonable picture will be that locaNMF is very robust for all of the 10 runs, but vanilla NMF will not be the case. To summarize, could the authors give a few more demonstrations where the outcomes of different runs for vanilla NMF and locaNMF can be shown? Guess the runs should be like vanilla NMF + random initiation, vanilla NMF + brain atlas initiation, and locaNMF.

(2) When using locaNMF to WFCI data, both spatial and temporal components are important. In practice, the variation of experimental data is large. Dependent on experimental conditions, there are some cases where neurons in one part of the brain region could be temporally silent during some period. Maybe it will be useful to discuss this point a bit. With the 2 datasets used in this study, could we say something about the temporal dynamics? Or to characterize the activity degree a bit across different brain regions? It could be possible that WFCI is generally good enough that neurons are always active in some way. Maybe it is difficult to find a proper exp. data, then a simulated data, like the one used in this study, could help by changing their temporal dynamics a bit to generate some imbalanced states.

(3) Similar to question (2), it seems that some data were not precisely aligned to the CCF, e.g. Fig 6 and 7. For example, the plot in Fig.6(D), if one sort the brain regions according to the similarity and plot after sorting. There are a few regions having a low index of similarity, for instance, RS part. Then for these less similar regions, what could be the reason? Because of their low active temporal dynamics? In another word, again, by simulated data, one can explore this in detail, I think. Then, one can get some guidelines that under which conditions, locaNMF may fail. These conditions may work together with the sensitivity of the hyperparameters. For example, different levels of localization threshold may depend on the temporal dynamics?

(4) line 573: "For runtime of LocaNMF on datasets of several sizes, see the Results section." It seems that there is no such part of the results in the text.

(5) Line 258k dPCs is not defined.

(6) Line 548 sNMF is not defined.

**Have all data underlying the figures and results presented in the manuscript been provided?**

Reviewer #1: None

Reviewer #2: No: simulation data are online, experimental datasets are not available

PLOS authors have the option to publish the peer review history of their article (what does this mean?). If published, this will include your full peer review and any attached files.

Reviewer #1: No

Reviewer #2: No

---

## [Decision Letter · Decision Letter 1]

17 Mar 2020

Dear Saxena,

We are pleased to inform you that your manuscript 'Localized semi-nonnegative matrix factorization (LocaNMF) of widefield calcium imaging data' has been provisionally accepted for publication in PLOS Computational Biology.

Best regards,

Jakob H Macke

Associate Editor

PLOS Computational Biology

Samuel Gershman

Deputy Editor

PLOS Computational Biology

Dear authors,

I have now read the revised version of the manuscript and the referee reports of the second round-- I am satisfied with the additional changes and analysis of sensitivity of initial conditions. Apologies for the delay.

A very minor comment from me:

"with temporal resolution limited only by the activity indicator and camera speeds.”— this is obviously true with respect to calcium, but sidesteps that fact that calcium dynamics themselves are much slower than voltage dynamics or spikes (which are often but not always the quantity of interest), so maybe add ‘calcium dynamics’ to this list.

Reviewer's Responses to Questions

**Comments to the Authors:**

Reviewer #1: The authors have addressed our comments.

Reviewer #2: Thanks for your revision. The authors addressed all of my concerns. In particular, the newly added figures provide a better comparison to other related methods. After correcting a potential error for similarity index, the evidence of using LocaNMF is more visible and clear in Fig.6 & 7 and Fig. SI 4, 5 and 6.

**Have all data underlying the figures and results presented in the manuscript been provided?**

Reviewer #1: None

Reviewer #2: No: Simulation data are online, experimental data are not available

PLOS authors have the option to publish the peer review history of their article (what does this mean?). If published, this will include your full peer review and any attached files.

Reviewer #1: No

Reviewer #2: Yes: Jian K. Liu

---

## [Editor Report · Acceptance letter]

1 Apr 2020

PCOMPBIOL-D-19-01682R1 

Localized semi-nonnegative matrix factorization (LocaNMF) of widefield calcium imaging data

Dear Dr Saxena,

I am pleased to inform you that your manuscript has been formally accepted for publication in PLOS Computational Biology. Your manuscript is now with our production department and you will be notified of the publication date in due course.

With kind regards,

Laura Mallard
